# The Involvement of the Endocannabinoid, Glutamatergic, and GABAergic Systems in PTSD

**DOI:** 10.3390/ijms26135929

**Published:** 2025-06-20

**Authors:** Anna Dorota Grzesińska

**Affiliations:** Collegium Medicum, Jan Dlugosz University in Czestochowa, ul. Waszyngtona 4/8, 42-200 Częstochowa, Poland; a.grzesinska@ujd.edu.pl

**Keywords:** endocannabinoid system, GABAergic system, glutamatergic system, post-traumatic stress disorder

## Abstract

Post-traumatic stress disorder (PTSD) is a debilitating mental health condition that develops in response to traumatic events. The endocannabinoid, glutamatergic, and GABAergic systems play crucial roles in the neurobiological mechanisms of PTSD. Both the endocannabinoid, glutamatergic, and GABAergic systems are involved in synaptic remodeling and neuronal differentiation, ensuring efficient information transmission in the brain. Their interplay influences motivation, behavior, sensory perception, pain regulation, and visual processing. Additionally, these systems regulate processes such as cellular proliferation, adhesion, apoptosis, and immune responses. This article explores the involvement of the endocannabinoid, glutamatergic, and GABAergic systems in PTSD pathogenesis. A literature review was conducted on studies examining the relationship between the endocannabinoid, glutamatergic, and GABAergic systems in PTSD. Relevant publications were sourced from the Web of Science and Scopus databases, covering research up to 29 February 2025. Neurobiological mechanisms underlying PTSD may share common pathways with other mental and somatic disorders, particularly those involving inflammatory processes. The identification of biomarkers is crucial for assessing PTSD risk and implementing targeted interventions to improve patient outcomes. A deeper understanding of these mechanisms could enhance therapeutic strategies, ultimately improving the quality of life for individuals affected by PTSD.

## 1. Introduction

Post-traumatic stress disorder (PTSD) is a debilitating dysfunction of the nervous system that arises from directly experiencing traumatic event(s) or witnessing them firsthand as they occur to others (Criterion A in the *Diagnostic and Statistical Manual of Mental Disorders, Fifth Edition* (DSM-5)) [1,2]. A hallmark of PTSD is the presence of recurring intrusive symptoms (Criterion B), in which the traumatic event is persistently re-experienced through distressing memories, nightmares, flashbacks, emotional distress triggered by trauma-related cues, and physical reactions such as excessive sweating or body tremors in response to reminders of the trauma. To cope, individuals with PTSD often engage in avoidance behaviors (Criterion C) [3,4]. This includes avoiding internal stimuli—such as distressing memories, thoughts, or emotions—as well as external reminders of the trauma, including specific people, places, conversations, activities, objects, or situations that elicit trauma-related distress [5]. In addition, PTSD is characterized by negative alterations in cognition and mood (Criterion D), which emerge or worsen following the traumatic event(s). These changes manifest in at least two of the following symptoms: difficulty recalling key aspects of the trauma; persistent, exaggerated negative beliefs about oneself, others, or the world; distorted perceptions of the trauma’s cause or consequences, leading to self-blame or blame toward others; ongoing negative emotional states; diminished interest in significant activities; feelings of detachment or estrangement from others; and a persistent inability to experience positive emotions [6,7]. Notably, individuals with PTSD may suppress traumatic memories during the day, leading to an emotional buildup that disrupts sleep at night [8,9,10]. Research suggests that the perception of threat—particularly the belief that one’s life was in danger—is a key predictor and maintainer of PTSD following trauma exposure [9,10,11]. An exaggerated sense of threat contributes to heightened vigilance and a persistent feeling of insecurity. Additionally, an impaired ability to recognize safety signals may serve as a biological marker, potentially contributing to the intergenerational transmission of PTSD [12,13,14]. PTSD is a functional disorder characterized by the body’s response to stress. Two primary systems are activated in the pathophysiology of the stress response: the sympathetic-adrenomedullary (SAM) system and the hypothalamic–pituitary–adrenal (HPA) axis [Figure 1]. The SAM system consists of the sympathetic part of the autonomic nervous system and the adrenal medulla. The HPA axis, on the other hand, is a neuroendocrine system that begins with the secretion of corticotropin-releasing hormone (CRH) by the paraventricular nucleus of the hypothalamus and culminates in the release of glucocorticoids by the adrenal cortex. The hypothalamus acts as a key integrator and regulator of the stress axis [15,16]. Autonomic and endocrine activation occurs through signals from both the internal environment and higher levels of the nervous system during the stress response. Activation of the sympathetic-adrenal system triggers the release of biogenic amines, such as adrenaline and noradrenaline, by the adrenal medulla cells. Additionally, significant amounts of noradrenaline are secreted from the extra-ganglionic neurons of the sympathetic system [17]. Catecholamines bind to adrenergic receptors, affecting internal organs and preparing the body to respond to stressors. The stress axis forms the limbic–hypothalamic–pituitary–adrenal axis through its connections to the hippocampus and amygdala, both components of the limbic system. The hormonal cascade of the HPA axis is initiated by CRH release from the paraventricular nucleus of the hypothalamus in response to stressors [18]. Beyond the hypothalamus, CRH also plays a crucial role in stress regulation, as CRH-containing neurons are distributed throughout various brain regions, including the frontal cortex, central nucleus of the amygdala, nucleus accumbens, gray matter, locus coeruleus (LC), raphe nuclei, and cingulate gyrus. This widespread distribution allows CRH to drive processes linked to anxiety behaviors [19]. A high concentration of catecholamines synthesized during a stress response significantly impairs prefrontal cortex (PFC) function. The PFC plays a vital role in regulating behavioral and emotional processes due to its extensive neuronal connections with the amygdala and locus coeruleus in the brainstem [20,21]. Stress-induced amygdala hyperactivity was also observed. Fluctuations in catecholamine levels—specifically noradrenaline (NA) and dopamine (DA)—either enhance or diminish PFC neuronal activity [22]. Under optimal arousal conditions, moderate catecholamine synthesis and secretion activate high-affinity α2A adrenergic receptors in the dendritic spines of the dorsolateral prefrontal cortex (dlPFC), supporting cognitive functioning. However, during stress, elevated catecholamine levels activate low-affinity α1 and β1 adrenergic receptors, weakening PFC function. This impairs cognitive processes and amplifies amygdala activity, which, in turn, stimulates the LC through CRH secretion, increasing LC neuron excitability and catecholamine release [23,24].

Nuclear factor kappa-light-chain-enhancer of activated B cells (NF-κB), a protein complex that regulates gene expression, plays a significant role in the inflammatory response in PTSD. NF-κB activation leads to the transcription of genes encoding pro-inflammatory cytokines. NF-κB signaling is involved in long-term memory formation and synaptic plasticity, which are processes that are disrupted in PTSD. The hippocampus, a brain region crucial for memory and stress response, shows altered NF-κB activity in individuals with PTSD.

The bed nucleus of the stria terminalis (BNST) is involved in the regulation of stress responses through its projections to the PVN of the hypothalamus, which caused the activation of the HPA axis and glucocorticoid release. Additionally, the BNST modulates the activity of the ventral tegmental area (VTA), a key dopaminergic nucleus implicated in motivational and affective processes. Neural responses within these interconnected regions exhibit temporally extended and delayed activation profiles, which can be quantitatively characterized using the Gaussian Delay Gaussian (GDG) model. The GDG model captures the timing dynamics of their interactions and individual variability in stress responses. This approach enables a deeper understanding of the neurobiological mechanisms underlying stress and anxiety disorders.

The hippocampus is essential for the negative feedback regulation of the HPA axis due to its high concentration of glucocorticoid receptors (GRs), which sense cortisol levels in the bloodstream and influence the hypothalamus to adjust the secretion of CRH. This feedback system is crucial for restoring balance during and after stress. In disorders such PTSD, this regulatory pathway becomes impaired. Prolonged exposure to stress and elevated glucocorticoids—especially during early life or traumatic experiences—can cause the hippocampus to shrink, weakening its control over the HPA axis. As a result, typical features of PTSD emerge, such as low cortisol levels and heightened sensitivity of glucocorticoid receptors.

The hypothalamic–pituitary–adrenal axis is a hormonal system that integrates structural and functional connections between the hypothalamus, pituitary gland, and adrenal gland. Through negative feedback mechanisms, it maintains a constant exchange of information [15,23]. The HPA axis is regulated by neural networks in the hypothalamus, hippocampus, prefrontal cortex, and amygdala. In PTSD, the amygdala becomes hyperactive, while the hippocampus and medial prefrontal cortex (mPFC) exhibit hypoactivity [19,22]. Other key structures involved in stress responses include the dorsal raphe nucleus (DRN), VTA, BNST. The VTA, a limbic nucleus in the midbrain, sends dopaminergic projections to the limbic system. The BNST plays a crucial role in the neural stress response, with the anterior BNST rich in gamma-aminobutric acid (GABA), adrenergic, androgen, vesicular glutamate transporter 3 (VGLUT3), and Pituitary adenylate cyclase-activating polypeptide type I receptor (PAC1 receptors), while the posterior BNST contains glutamate and kainate receptors. The BNST is also a primary target of the DRN’s serotonergic projections. PTSD is associated with imbalances in multiple neurotransmitters, including glutamate, GABA, dopamine, and acetylcholine (ACh) [17,20,24]. The HPA axis is essential for both stress adaptation and response, initiating a cascade of changes through the release of CRH and arginine vasopressin (AVP) from the paraventricular nucleus (PVN) into the pituitary gland’s portal circulation. These hormones stimulate the secretion of adrenocorticotropic hormone (ACTH), which then reaches the adrenal cortex via the bloodstream, triggering the release of glucocorticoids. Glucocorticoids play a key role in modulating neuroinflammation, particularly in the context of PTSD, through their impact on emotional and cognitive processes. Cortisol, produced by the adrenal cortex in response to stress via activation of the HPA axis, plays an important role in modulating immune responses and suppressing inflammation. Cortisol exerts immunosuppressive and anti-inflammatory effects by inhibiting the production of pro-inflammatory cytokines such as Interleukin-6 (IL-6), inducing anti-inflammatory cytokines like Interleuki Interleukin 6n-10 (IL-10), reducing the activity of macrophages and T lymphocytes, and suppressing the activation of the NF-κB pathway, which regulates the expression of inflammatory genes. An increase in cortisol in response to stress inhibits the inflammatory response. However, prolonged stress may lead to hypocortisolism in PTSD and an exacerbation of the inflammatory state (Figure 1).

Research shows that brain regions such as the amygdala and PFC are central to PTSD pathophysiology [25,26,27]. The amygdala is responsible for recognizing threats in the environment and triggering alertness. In PTSD, the amygdala becomes hyperactive, exhibiting heightened metabolic activity during symptom flare-ups and exaggerated responses to trauma memories, even in the absence of direct stimuli [17,18,23]. In addition to the amygdala, subregions of the PFC, particularly the ventromedial prefrontal cortex (vmPFC), are implicated in PTSD development and maintenance [28]. Studies suggest that the vmPFC is underactive in individuals with PTSD, especially when processing trauma-related cues or attempting to regulate and extinguish fear responses [29,30].

In a study by Helpman et al. [29], it was shown that individuals with more severe PTSD symptoms exhibited significantly reduced activity in the vmPFC during fear extinction. PTSD patients also showed a reduction in rostral anterior cingulate cortex (rACC) activation from pre- to post-treatment during extinction recall, along with increased functional connectivity between the rACC, vmPFC, and the subgenual anterior cingulate cortex (sgACC). A meta-analysis of functional magnetic resonace imaging (fMRI) studies by Fullana et al. confirmed that the vmPFC is a key brain region involved in extinction processes, and its reduced activity may be associated with difficulties in extinguishing fear responses in individuals with anxiety disorders [30]. Impaired vmPFC recruitment during fear extinction is linked to the difficulties in extinguishing conditioned fear responses observed in PTSD. Reduced vmPFC activity inversely correlates with PTSD symptom severity, and there is a well-documented negative relationship between vmPFC and amygdala activation—hyperactivity of the amygdala in PTSD often corresponds with hypoactivity of the vmPFC [29,30,31,32]. The biological underpinnings of PTSD remain a critical area of scientific inquiry, as hormonal, immunological, and epigenetic dysregulation not only contribute to mental health disorders but also lay the foundation for various physical illnesses [15,27].

This publication provides an overview of selected studies exploring the relationship among the endocannabinoid, glutamatergic, and GABAergic systems in PTSD.

The study utilized the PubMed and Web of Science databases, including articles published up to 26 March 2025. It is a review paper that establishes the substantive foundation for research conducted under the Era-Net Neuron 20/2019 grant. This article focuses on analyzing the roles of the endocannabinoid, glutamatergic, and GABAergic systems in the trauma process. The grant encompasses long-term scientific observations aimed at mapping and interrogating top-down control of the memory engram in post-traumatic stress disorder. Future studies, to be presented in subsequent publications, will provide insights into the psychological, biochemical, genetic, and neuroimaging aspects of PTSD.

## 2. Alterations in the Stress-Buffering Endocannabinoid System (ECS)

The endocannabinoid system plays a crucial role in maintaining homeostasis within the human body, aiding adaptation to both physiological and pathophysiological changes [Figure 2].

In response to stress, the HPA axis is activated, leading to the release of cortisol. Excessive cortisol levels can damage neurons, especially in the hippocampus—a structure responsible for memory and learning. This can result in decreased neuronal plasticity. Moreover, the amygdala becomes hyperreactive, which increases anxiety and emotional disturbances. This may also lead to difficulties in decision-making and impulse control. In contrast, the endocannabinoid system plays a protective role by “silencing” fear memories. Endocannabinoids have properties that reduce anxiety, pain, and depression, and they also improve sleep and appetite.

These adaptive actions include regulating gene expression—particularly genes responsible for silencing mechanisms (blocking transcription or translation)—enhancing beneficial immune responses, regulating metabolism and energy balance, and supporting brain–gut axis function. Furthermore, the ECS influences synaptic plasticity, learning processes, pain perception, stress and emotional regulation, and sleep cycles [33,34]. In response to acute stress, the ECS becomes deactivated, while in chronic stress, it helps mitigate the adverse effects of dysregulated adrenal cortex hormones. Notably, the ECS is involved in fear extinction processes by reducing amygdala reactivity, which plays a pivotal role in anxiety regulation [35].

The ECS comprises cannabinoid receptors (CB1 and CB2), which belong to the family of 7-transmembrane receptors, and their ligands. Additionally, enzymes involved in ligand synthesis and degradation—primarily those derived from arachidonic acid—regulate this system. Ligand interaction with CB1 and CB2 receptors alters their spatial configuration by activating G-proteins, inhibiting adenylate cyclase, and stimulating the mitogen-activated protein kinase (MAPK) pathway. These processes activate potassium channels, causing potassium efflux from the cell [36]. Moreover, the activation of CB receptors inhibits N- and P/Q-type calcium channels, reducing calcium influx into the cell. This mechanism operates independently of cyclic adenosine monophosphate (cAMP) concentration. The ECS is closely linked to the proper functioning of both the autonomic and central nervous systems, facilitating synaptic transmission and the formation of neuronal connections [37,38]. Studies show that the ECS contributes to the formation and functioning of neural networks in brain regions such as the hippocampus and cerebellum [39,40,41]. Endocannabinoids also play a role in remodeling neuronal connections and supporting neuronal differentiation. Additionally, cannabinoid receptors in the brain interact with key neurotransmitters, including serotonin, glutamate, and GABA, which may contribute to the presynaptic inhibition of neurotransmitter release. Specifically, CB1R activation in GABAergic neurons were shown to reduce active coping strategies during fear conditioning in response to stress stimuli [42]. In trauma-induced stress responses, the body attempts to restore homeostasis by activating survival mechanisms, which include autonomic responses via the sympathetic nervous system and neuroendocrine responses linked to the HPA axis [43]. Research suggests that the ECS plays a critical role in modulating this neuroendocrine response [33,34,35]. CB1R receptors are predominantly located in the central nervous system—within the cerebral cortex, amygdala, hippocampus, and basal ganglia—whereas CB2R receptors are mainly distributed around the nervous and immune systems. Both receptors are presynaptic G-protein-coupled receptors found at the axon terminals of GABAergic, glutamatergic, and monoaminergic neurons. In studies conducted on human material, the authors demonstrated that the activation of cannabinoid receptors is associated with the endogenous synthesis of endocannabinoids (eCBs) in postsynaptic neurons, followed by retrograde signaling to presynaptic CB receptors. CB1R activation inhibits neurotransmitter release—such as glutamate or GABA—at the synapse, modulating both inhibitory and excitatory synaptic activity [39,43]. Studies using animal models showed that CB1R dysfunction correlates with heightened anxiety phenotypes, while CB1R agonists help reduce trauma-related anxiety behaviors [44,45,46,47,48,49]. Several biomarkers linked to the ECS, PTSD, and sleep disorders were identified in the literature, including NOD-like-receptor containing a Pyrin domain 1 (NLRP1), caspase-1, N-methyl-D-aspartate (NMDA) receptor, and GABA [50,51,52].

The ECS and HPA axis interact closely in the pathogenesis of PTSD and the fear memory process. The ECS serves as a crucial modulator of stress, particularly during early-life trauma [10,34]. Research on past trauma across various age groups highlights that early-life trauma is reflected in altered concentrations of eCBs and CB1R ligands [41,46]. Studies in healthy individuals and animals showed increased eCB ligand levels and reduced CB1R expression [37,38]. Conversely, in trauma-exposed individuals, eCB ligand levels tend to decrease, while CB1R expression increases [38]. Other studies reported elevated levels of eCB ligands [1-arachidonoylglycerol (1-AG), 2-arachidonoylglycerol (2-AG)] or anandamide (AEA) and altered CB1R expression in key brain areas, including the hippocampus, amygdala, and mPFC [53].

The endocannabinoid system is crucial for memory formation and for linking experiences to specific behavioral responses. In the amygdala, certain interneurons contain cannabinoid receptors that help regulate the size and specificity of memory traces (engrams). However, excessive endocannabinoid release triggered by stress can disrupt the function of these interneurons, leading to distorted memory engrams and various memory disorders such as hypermnesia, amnesia, or flashbacks.

Furthermore, stress-induced overactivation of the endocannabinoid system may contribute to the formation of generalized fear memories. By selectively blocking cannabinoid receptors on specific interneurons, it may be possible to prevent one of the most debilitating symptoms of PTSD. Top-down control in PTSD refers to the influence of higher brain structures (e.g., the prefrontal cortex) on lower regions such as the amygdala and hippocampus. In PTSD, top-down regulation includes the suppression of emotional responses via the mPFC, modulation of emotional memory reactivity (including that of traumatic engrams), and regulation of fear extinction processes.

## 3. The Glutamatergic System

Glutamatergic neurons represent the largest group in the central nervous system (CNS), with glutamate (glutamic acid) serving as the primary excitatory neurotransmitter. Glutamate plays a crucial role in five main glutamatergic pathways: the descending corticobrainstem pathway, the descending corticostriatal pathway, the ascending and descending corticothalamic pathways, and the corticocortical pathways that connect pyramidal cells [54]. Glutamate is synthesized from glutamine in glial cells, primarily via three mechanisms. One involves the reduced form of nicotinamide adenine dinucleotide phosphate (NADPH) and the enzyme glutamate dehydrogenase, where alpha-ketoglutarate serves as the substrate. Another pathway involves the transfer of an amino group from an L-amino acid to alpha-ketoglutarate via glutamate aminotransferase. The third mechanism involves the hydrolysis of glutamine by glutaminase, with the aid of zinc [55]. Once produced, glutamate is transferred to synaptic vesicles for storage and is released into the synaptic cleft during neuronal depolarization. The reuptake of glutamate is mediated by excitatory amino acid transporters (EAATs), which are located in astroglia and microglia, where glutamate is converted back into glutamine and transported back to presynaptic terminals. Glutamate interacts with several membrane and metabotropic receptors, including the NMDA receptor, the α-amino-3-hydroxy-5-methyl-4-isoxazolepropionic acid receptor (AMPAR), and the kainate receptors (GRIK) [56]. Ionic NMDA receptors, mainly found in the hippocampus, can contain the following subunits: NR1, NR2, and NR3. These subunits serve as binding sites for glycine, glutamate, or D-serine. Interestingly, the metabolism of D-serine is associated with the activity of enzymes such as D-amino acid oxidase (DAO), which catalyzes the conversion of D-serine to hydroxypyruvate, and its activator, D-amino acid oxidase activator (DAOA). The NMDA receptor can have either an activating or inhibiting effect, depending on the subunit type. When NR3 is activated, glycine exerts an activating effect, whereas D-serine has an inhibiting effect. In contrast, for the NR2 subunit, both amino acids exhibit an activating effect [57]. It is believed that stimulated postsynaptic ionotropic receptors contribute to the opening of ion channels and the influx of calcium ions into the cell, which triggers an action potential. This process activates intracellular enzymes such as protein kinase A (PKA), mitogen-activated protein kinases (MAPK), and calmodulin. Furthermore, at these sites, beyond receptor saturation by agonists (glutamic acid, aspartic acid), protein activity can be modulated. Research highlights the importance of strategic neurotransmitter binding sites, such as the glycine binding site, which facilitates ion channel opening, and the phencyclidine (PCP) binding site, which blocks the NMDA receptor ion channel [47,56]. Additionally, ions like Mg^2+^ and Zn^2+^ play crucial roles in the biochemical process known as kindling. Studies confirm that magnesium ions (Mg^2+^) block ion channels, while zinc (Zn^2+^) inhibits NMDA receptor activation [50,52]. Scientific observations emphasize the role of free radicals in modulating redox potential, thereby influencing oxidation and reduction processes in the brain [54]. Furthermore, the activation of NMDA receptors in the brain stimulates free radical production, which in turn affects steroid binding [58]. Additionally, specific magnesium (Mg^2+^) and zinc (Zn^2+^) ions localized on the receptor can prevent the flow of other cations through the open ion channel. This process depends on the intracellular movement of sodium (Na^+^), potassium (K^+^), and calcium (Ca^2+^) ions during cell depolarization, which displaces Mg^2+^ and Zn^2+^ from the channel. The influx of Ca^2+^ through NMDA receptors is particularly significant for synaptic plasticity, cognitive modulation, anxiety formation, pain perception, and learning mechanisms [59]. Importantly, NMDA receptor blockade can reduce uncontrolled calcium ion influx into neurons, preventing cell necrosis. Moreover, NMDA activity can be inhibited by various substances, such as ketamine. Excessive NMDA receptor activation by NMDA agonists elevates cytosolic zinc or calcium concentrations, which may lead to cell necrosis. The overactivation of NMDA receptors is thought to contribute to mental illnesses such as anxiety and depressive disorders by inducing neuronal atrophy, disrupting glutamate neurotransmission, and inhibiting hippocampal neurogenesis. Thus, NMDA receptor inhibition is considered a potential treatment for diseases linked to excessive glutamate system activity. Cannabinoids can counteract this process through CB1 receptor activation, a mechanism involving histidine triad nucleotide-binding protein 1 (HINT-1). The neurotoxic effects of excessive NMDA receptor activation are mitigated by inhibiting nitric oxide synthase (nNOS) expression and activating ascorbate. Methamphetamine has been noted for its toxic impact on the brain, reducing damage and inhibiting astroglia via a CB2 receptor-dependent mechanism for phytocannabinoid tetrahydrocannabinol (THC) (delta-9-THC). Ongoing debate surrounds the potential inhibition of receptor activation and excitotoxicity [60].

Excitotoxicity occurs when the prolonged exposure of nerve cells to glutamate leads to excessive NMDA receptor stimulation, resulting in a significant rise in intracellular calcium concentration and, consequently, cell death. This phenomenon is particularly relevant in cases of blood extravasation and cerebral hypoxia, making it crucial for managing the progression of mental disorders associated with post-traumatic stress disorder. Blocking NMDA receptors could, therefore, play a key role in treating such conditions [61,62]. As mentioned earlier, stimulated NMDA receptors are essential for neuronal plasticity and learning processes. These states are linked to gene expression regulation, dendritic and axonal spatial modeling, neural network organization, synaptic transmission, cellular signal modification, and neurotransmitter secretion changes. Synaptic plasticity begins with neurotransmitter stimulation and neuronal connection activation, with neuronal conduction triggered by receptor phosphorylation. Additionally, the scientific literature describes the formation of new nerve endings during synaptic pruning, a critical mechanism for strengthening neuronal connections and reconstructing damaged pathways [37,47]. Proteins involved in transcription and translation participate in this process, which also occurs in apoptosis mechanisms. Neural networks undergo modifications by selecting the most functional neurons, with neuronal reprogramming forming the basis for neuronal plasticity mechanisms underlying memory and learning. Notably, interactions between neighboring neurons align with Hebb’s rule, which states that the simultaneous stimulation of two neurons enhances synaptic transmission efficiency. Repetitive neuronal stimulation contributes to the long-term potentiation (LTP) of synaptic excitation, whereas weakened stimuli lead to long-term depression (LTD). NMDA receptors have been implicated in LTP-dependent synaptic excitation, particularly in the CA3 and CA1 regions of the hippocampus and pyramidal cells. A sufficiently strong stimulus in CA1 neurons induces postsynaptic depolarization with NMDA receptor involvement [63,64]. Another important glutamatergic system receptor is the AMPAR, an ionotropic receptor crucial for fast synaptic transmission and long-term potentiation. AMPA activation causes a substantial Na^+^ influx and a slight K^+^ outflow, leading to postsynaptic neuron depolarization. AMPARs, along with NMDA receptors, are found in excitatory synapses, where they work together to open calcium channels in response to stimuli. NMDA receptors operate more slowly and sustain longer activity, whereas AMPARs act more rapidly and briefly. Ca^2+^ ion entry activates protein kinases and promotes the phosphorylation of both AMPA and NMDA receptors. Additionally, protein kinases such as PKA influence the transcription factor cAMP response element-binding protein (CREB), which regulates gene expression related to synaptic plasticity, trophic factors (e.g., brain-derived neurotrophic factor, BDNF), and membrane receptors (e.g., GluA1 subunits of the AMPAR). There are four genes that encode for AMPA AMPARs (GRIA1 to GRIA4), and each AMPAR is composed of four subunits, which can form either homomeric or heteromeric assemblies of GluA1 to GluA4. Most AMPARs contain at least one GluA2 subunit. After transcription, the GluA2 subunit undergoes ribonucleic acid (RNA) editing, in which the codon for a glutamine residue in the ion channel pore is replaced with one that codes for arginine [65]. This RNA editing is a crucial step in quality control. Unedited GluA2 is retained in the endoplasmic reticulum. The larger arginine residue in the pore region of the channel restricts the flow of Na^+^ and K^+^ ions and prevents the entry of divalent ions into the cell, making AMPARs that include GluA2 calcium-impermeable. AMPARs typically carry inward currents at negative potentials and outward currents at positive potentials, with a reversal potential at 0 mV. For receptors containing GluA2, the current–voltage relationship is linear and symmetrical. However, for receptors that lack GluA2, such as GluA1 homomeric or GluA1/3 heteromeric channels, the current–voltage relationship exhibits a nonlinear, voltage-dependent behavior. These GluA2-lacking AMPARs have a glutamine residue in the pore region instead of arginine, resulting in a high conductance for sodium and even permeability to calcium.

Additionally, negatively charged endogenous polyamines can bind near the cytoplasmic end of the pore, inhibiting the channels at positive potentials. As a result, GluA2-lacking AMPARs exhibit an inward-rectifying current–voltage relationship, meaning they conduct current more easily into the cell than out. These calcium-permeable AMPARs are primarily found on many GABAergic neurons [66]. Another receptor associated with the glutamatergic system is the kainate receptor (KAR), primarily located in the hypothalamus, thalamus, caudate nucleus, and hippocampus. KARs are a type of ionotropic glutamate receptor involved in excitatory neurotransmission, synaptic plasticity, and the regulation of neurotransmitter release [67]. These receptors play key roles in both contextual and cued fear responses. KARs consist of subunits GluK1 to GluK5, and they exert both presynaptic and postsynaptic effects on glutamatergic and GABAergic synaptic transmission. Presynaptic KARs are involved in neurotransmitter release, while postsynaptic KARs regulate neuronal excitability. In clinical research, silencing the GluK5 subunit of kainate receptors showed promise in treating drug-resistant epilepsy. Furthermore, disruptions in KAR function are linked to various neurological and psychiatric disorders, including PTSD, anxiety disorders, epilepsy, and depression. The functional diversity of kainate receptors is enhanced by RNA editing and alternative splicing. For instance, GluK1-containing KARs, predominantly found in the hippocampus, cortical interneurons, Purkinje cells, and sensory neurons, undergo alternative splicing, producing isoforms such as GluK1-a, GluK1-b, GluK1-c, and GluK1-d [68].

Another group of receptors, metabotropic glutamate receptors (mGluRs), are associated with secondary messenger systems and are coupled to Guanosine Triphosphate -binding (GTP-binding) proteins. These receptors, located pre- and postsynaptically on glutamatergic and GABAergic synapses, are involved in regulating phosphatidylinositol metabolism and intracellular nucleotide synthesis. The mGluRs are classified into three groups: Group I (mGluR1, mGluR5), Group II (mGluR2, mGluR3), and Group III (mGluR4, mGluR6, mGluR7, mGluR8). Group I receptors, including mGluR1 and mGluR5, activate phospholipase C and lead to the synthesis of inositol-1,4,5-triphosphate (IP3) and diacylglycerol (DAG), which in turn activates protein kinase C (PKC) and releases intracellular calcium. These receptors also engage with adenylate cyclase and the extracellular signal-regulated kinases, mitogen-activated protein (ERK MAP) kinase pathways, influencing processes like cell differentiation and apoptosis and contributing to glutamate-induced neurotoxicity and pain perception. In contrast, Groups II and III receptors inhibit glutamate release. The mGluR3, in particular, is involved in inhibiting adenylate cyclase activity via G protein binding [69].

The mGlu5 receptor is primarily localized postsynaptically and on glial cells [70]. It regulates glutamatergic neurotransmission, mediates synaptic plasticity, and plays a key role in cognitive functions such as fear learning, memory formation, and emotion regulation. Studies identified mGlu5 dysfunction in several clinical populations, including individuals with PTSD [66]. Notably, evidence suggests that mGlu5 is central to the acquisition and consolidation of trauma-related memories (i.e., fear conditioning) and is implicated in PTSD-like behaviors in animal models, such as freezing, contextual fear memory, and fear generalization. Human studies using positron emission tomography (PET) revealed higher mGlu5 availability in individuals with PTSD compared to healthy adults and those with major depressive disorder (MDD) [66,67]. Moreover, the degree of mGlu5 dysregulation correlates with symptom severity, including avoidance, anxiety, tension, and suicidal ideation. Preclinical research on PTSD models indicates that post-traumatic stress alters mGlu5 expression in the hippocampus (HIP) and prefrontal cortex [71,72]. Notably, mGlu5 expression is often accompanied by changes in BDNF levels. Additionally, BDNF/tropomyosin receptor kinase B (TrkB) signaling contributes to neuroplasticity and exhibits antidepressant effects by regulating the expression of glutamate transporter-1 (GLT-1). The mGlu receptors play a crucial role in excitatory neurotransmission, intercellular communication, synaptic plasticity, metaplasticity, neuronal fine-tuning, and gene expression [73]. Given their involvement in these fundamental processes, it is unsurprising that mGlu receptors help sustain cellular activity underlying many cognitive functions.

Metabotropic glutamate receptors act as regulators of neuronal activity and synaptic plasticity, making them crucial for brain functions related to cognition and memory. These receptors activate G proteins, which trigger various intracellular signaling pathways, enabling the long-term modulation of neuronal excitability, neurotransmitter release (not only glutamate but also GABA, dopamine, and acetylcholine), and gene expression associated with synaptic plasticity. The mGluRs are central to processes such as LTP, associated with learning and memory, and LTD, which modulates synaptic strength—both of which are essential for neuroplasticity.

The glutamatergic system, acting via the neurotransmitter glutamate, plays a significant role in the pathogenesis of PTSD. Dysregulation of this system, particularly in the hippocampus, may be linked to PTSD symptoms such as memory disturbances, intrusive traumatic memories, and difficulties in emotional regulation. Moreover, therapies targeting the glutamatergic system can help alleviate PTSD symptoms and improve the quality of life for affected individuals.

## 4. The GABAergic System

Gamma-aminobutyric acid is a non-proteinogenic amino acid that acts as the principal inhibitory neurotransmitter in the CNS. It plays a crucial role in regulating neuronal excitability by increasing the threshold for activation and reducing synaptic conductivity, thereby weakening the transmission of stimuli. GABA is synthesized from glutamate through the action of the enzyme glutamic acid decarboxylase (GAD), with pyridoxal phosphate acting as a cofactor. In clinical contexts such as PTSD, ischemia, and hypoxia, GABA levels can decrease within neurons, and GAD activity can be impaired, contributing to an imbalance in the excitatory–inhibitory signaling. The metabolism of GABA within neurons is regulated by GABA transaminase (GABA-T), which converts GABA into succinic semialdehyde, subsequently entering the Krebs cycle and eventually being converted back to glutamate. This conversion process, known as the “GABA shunt”, involves GAD, GABA-T, and succinate dehydrogenase, contributing to metabolic flux. The importance of homocarnosine—a compound formed by GABA and histidine—was also highlighted for its immunomodulatory and antioxidant functions. However, disturbances in GABA and glutamate metabolism can result in excitotoxicity, a process that underlies the pathophysiology of various CNS disorders. Research indicates that GABA and glutamate are not entirely distinct but interact within specific neuronal populations.

Moreover, approximately 80% of the glutamate content in glutamatergic neurons is found within these cells, whereas in GABAergic neurons, it accounts for only about 10% [74]. Research indicates the simultaneous presence of both GABA and glutamate in granule cells of the dentate gyrus, brainstem neurons, and motoneurons [50,51]. Additionally, studies showed that GABAergic interneurons in the hippocampus and neocortex can also release glutamate [70,75]. Furthermore, the glutamate reuptake of excitatory amino-acid transporter 4 (EAAT4) and excitatory amino-acid transporter 5 (EAAT5) was identified on GABAergic Purkinje neurons in the cerebellum and retina [76]. A study by Dong et al. [77] found that the simultaneous release of glutamate and GABA in epileptic patients contributes to homeostatic control, counteracting excitotoxicity. These clinical findings highlight the intricate interplay between the GABAergic and glutamatergic systems, fostering hope for the development of novel therapeutic agents capable of regulating neurological homeostasis [Figure 3]. Conversely, dysfunctions within these neurotransmitter systems may trigger a pathological cascade in the HPA axis, which plays a key role in the stress response. Interestingly, clinical observations suggest the rhythmic filling of synaptic vesicles with either glutamate or GABA. This process is mediated by vesicular transport proteins, such as vesicular GABA transporter (VGLUT1, VGLUT2, and VGLUT3) for glutamate and VGAT, also known as vesicular inhibitory amino acid transporter (VIAAT), for GABA. The activity of these transporters is dependent on adenosine triphosphate (ATP), sodium ions, and the proton pump. VGLUT1 and VGLUT2 are particularly associated with synapses in the hippocampus and neostriatum, where they interact with phosphoproteins in synaptic vesicles. This interaction is crucial for neurotransmission and the formation of neuronal connections [78]. A lack of activity in these regions leads to decreased glutamate release, triggering a compensatory mechanism that enhances GABA release at neuronal terminals. Notably, both glutamate and GABA are released into the synaptic cleft through two distinct mechanisms: Ca^2+^-dependent vesicular exocytosis, regulated by ATP, and Ca^2+^-independent release via the reversal of reuptake transporters. The release of neurotransmitters from presynaptic terminals is further modulated by receptor activity, underscoring the complex interplay between the two amino acid systems [79]. As previously mentioned, glutamate and GABA form a metabolic neurotransmitter system that influences excitatory postsynaptic potentials (EPSPs) and inhibitory postsynaptic potentials (IPSPs) via ionotropic receptors. The summation of neuronal discharges determines overall neuronal depolarization. In the neocortex, for example, modulatory presynaptic heteroreceptors on glutamatergic terminals—such as GABA_A_ receptors (GABA_A_Rs)—can inhibit glutamate release by reducing Ca^2+^-induced depolarization. This inhibition is linked to the activation of voltage-gated L- and R-type calcium channels and an increase in the protein kinase II [calcium/calmodulin-stimulated protein kinase II (CaMKII)]-mediated phosphorylation of synapsin I in conjunction with the K^+^/Cl^−^ cotransporter potassium-chloride transporter member-5 (KCC2) [80]. Additionally, in the hippocampus, the stimulation of presynaptic NMDA receptors enhances GABA release via a nitric oxide (NO)/cyclic GMP (cGMP)-dependent mechanism [81]. In addition, it is worth noting that mGluRs and GABA_B_ receptors (GABA_B_Rs) belong to the class C family of G protein-coupled receptors. This classification suggests that intracellular factors such as phospholipase C, adenylate cyclase, and ion channels play a role in their regulatory processes.

An interesting observation is that presynaptic GABA_B_ autoreceptors in hippocampal afferent fibers can be stimulated by a theta rhythm (5 Hz), which is associated with attentional focus. This stimulation leads to decreased GABA release and increased excitatory transmission in the hippocampal CA1 field, contributing to LTP, a key mechanism in memory consolidation. Additionally, GABA_B_ receptors (GABA_B_Rs) on presynaptic glutamatergic neurons were implicated in the regulation of LTD induced by AMPA and mGluR1 receptors. Activation of GABA_B_Rs engages G proteins, triggering intracellular mGluR1 signaling pathways involved in LTD [82]. Moreover, studies showed that GABA release can be either potentiated or attenuated by mGluR stimulation, with the effect depending on receptor subtype and exposure time [50,51]. The interplay between the GABAergic and glutamatergic systems is further emphasized by the presence of mGluR1 on presynaptic GABAergic terminals. The activation of these receptors inhibits calcium channel activity while activating potassium channels, ultimately reducing GABA release. Of particular clinical relevance are ongoing studies on cannabinoids and their role in modulating neurotransmission [Figure 3]. Notably, the activation of postsynaptic mGluR1 contributes to the release of endogenous cannabinoids from pyramidal cells in the hippocampal CA1 field [83]. These cannabinoids act by mimicking secondary messenger systems, activating CB1 receptors on presynaptic interneuron terminals to inhibit GABA release. Research by Singewald et al. suggests that CB1 receptors located on GABAergic presynaptic terminals interact with glutamatergic signaling in pain modulation [84]. Additionally, Xie et al. proposed that CB1 receptors may play a role in inhibiting GABAergic transmission following mGluR5 stimulation [46]. These findings have significant therapeutic implications, as ongoing research suggests potential applications of glutamatergic and GABAergic receptor modulators in the treatment of conditions such as PTSD and depression [85]. Furthermore, the neuromodulatory interplay between glutamate and GABA presents opportunities for pharmacological interventions targeting functional memory disorders, including those of vascular or neurodegenerative origin. Scientific evidence indicates that GABA_B_ autoreceptor stimulation influences LTP induction, a process also modulated by mGluR1 activation [86,87]. Alterations in the balance between GABAergic and glutamatergic signaling can enhance or impair learning and memory, underscoring the potential of pharmacological agents targeting these neurotransmitter systems. Pharmacological intervention targeting the GABA/glutamate imbalance is particularly beneficial in counteracting the overactivity of the glutamatergic system, which underlies excitotoxicity and contributes to neurodegenerative brain changes. Recent studies highlighted the therapeutic importance of restoring balance between glutamate and GABA following cerebral hypoxia [51,88]. Hypoxic events inevitably trigger a cascade of biochemical processes, including the decreased activity of ion channels (K^+^, Na^+^, and Ca^2+^), membrane hyperpolarization, and subsequent depolarization, leading to excessive glutamate release and excitotoxicity. As is well known, prolonged neuronal dysfunction can ultimately result in neuronal apoptosis [89]. Notably, these alterations—combined with increased glutamate synthesis and impaired glutamate release into the synaptic cleft—can lead to elevated intracellular glutamate concentrations and NMDA receptor hypofunction. This dysregulation in the interaction between the GABAergic and glutamatergic systems may further contribute to diminished brain receptor sensitivity, ultimately disrupting the dopaminergic system [47,50]. Such neurochemical imbalances have been implicated in the pathophysiology of various psychiatric disorders, including PTSD (classified under Axis I as an anxiety disorder), bipolar disorder, and schizophrenia (classified under Axis II). Additionally, other studies highlight the role of metalloproteinases, particularly matrix metalloproteinase-2 (MMP-2) and matrix metalloproteinase-9 (MMP-9), in synaptic plasticity. These proteolytic enzymes, regulated by the glutamatergic and cannabinoid systems, play crucial roles in physiological processes such as tissue repair. MMP-2 is primarily localized in astrocytes, whereas MMP-9 is predominantly found in neurons and dendrites. Notably, increased glutamate release and the activation of postsynaptic NMDA and AMPARs during long-term potentiation enhance the expression and activity of metalloproteinases. Given their involvement in synaptic remodeling, these enzymes are also implicated in the pathogenesis of various neurodegenerative diseases. Furthermore, Wu L. et al. described the interaction between GABA_B_ receptor (GABA_B_R) stimulation and Group I mGluR antagonists in the rat hippocampus, with evidence suggesting the involvement of MMP-2 and MMP-9 in these processes [90]. These findings underscore the complex interplay between the GABAergic and glutamatergic systems in regulating proteolytic enzyme activity, pointing to an additional level of brain function modulation [91].

In conclusion, the regulation of GABA and glutamate release is a crucial mechanism for maintaining the balance between the glutamatergic and GABAergic systems in the brain. Various factors can disrupt this equilibrium, triggering a cascade of reactions that impair neuronal function and contribute to CNS damage.

The GABAergic system also plays an important role in restoring the neurohormonal balance in PTSD. Increased GABAergic activity contributes to improved sleep quality—often disrupted in PTSD—and reduces anxiety and negative emotions associated with traumatic experiences.

## 5. Conclusions

A deeper understanding of the interaction between the cannabinoid and glutamatergic systems is essential for developing targeted therapeutic strategies for post-traumatic stress. Both systems play a critical role in modulating amygdala reactivity, influencing mechanisms of anxiety extinction and sensitization. Future advancements are likely to refine our ability to regulate the excitability of these systems with greater precision. Emerging techniques such as optogenetics, designer receptors exclusively activated by designer drugs (DREADDs), inhibitory glutamate receptors responsive to excessive glutamate spillover, and activity-dependent promoters targeting gene expression in hyperactive neurons hold significant promise for therapeutic intervention.

## Figures and Tables

**Figure 1 ijms-26-05929-f001:**
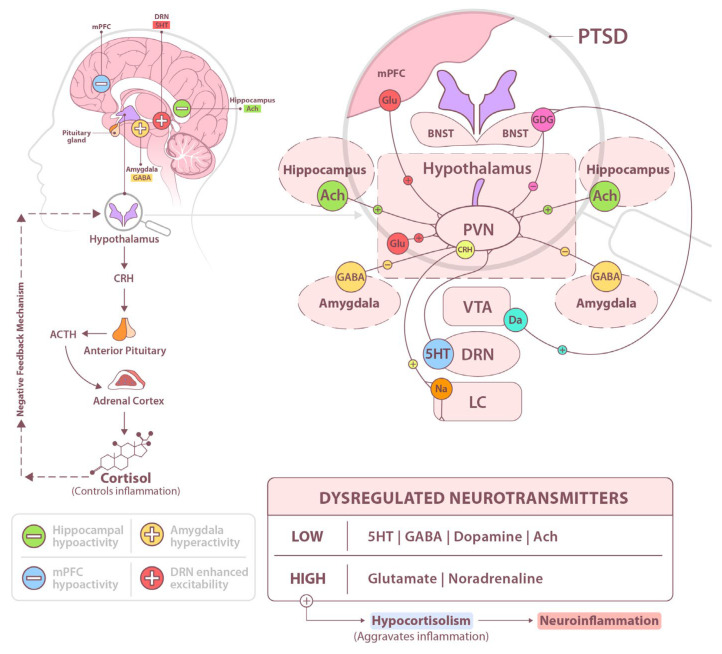
Neurohormonal regulation in relation to the hypothalamic–pituitary–adrenal (HPA) axis. In response to stress, the HPA axis is activated. This process is initiated in the hypothalamus, where neurons of the paraventricular nucleus (PVN) synthesize and release corticotropin-releasing hormone and arginine vasopressin (AVP). CRH stimulates the anterior pituitary gland, promoting the secretion of adrenocorticotropic hormone (ACTH). ACTH acts on the adrenal cortex, inducing the synthesis and release of glucocorticoids, primarily cortisol. Cortisol plays a pivotal role in modulating immune responses and influencing cognitive and emotional processes. Furthermore, through a negative feedback mechanism, cortisol inhibits activation of the HPA axis. In the context of post-traumatic stress disorder (PTSD), dysregulation of the HPA axis is frequently observed, typically manifesting as hypocortisolism, enhanced glucocorticoid receptor sensitivity, and impaired negative feedback control. This dysfunctional feedback mechanism is associated with anxiety symptoms, sleep disturbances, hyperarousal, and deficits in emotional regulation. These changes are linked to reduced levels of serotonin (5-HT) gamma-aminobutric acid (GABA), dopamine (DA), and acetylcholine (ACh) and increased glutamate and noradrenaline levels.

**Figure 2 ijms-26-05929-f002:**
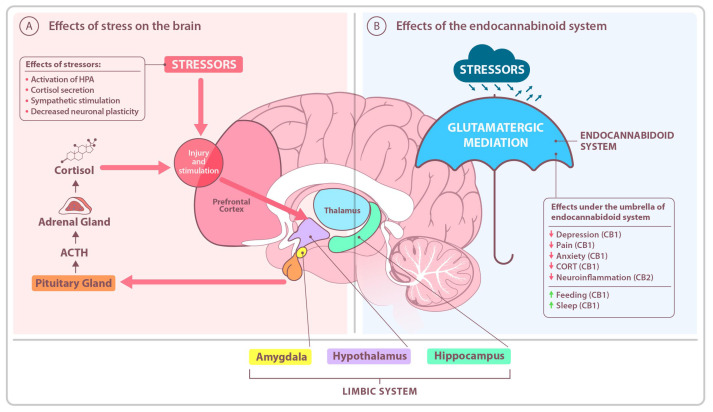
Effects of stress on the brain and the protective role of the endocannabinoid system. (**A**). Stress affects the central nervous system by activating key brain structures, including the limbic system (comprising the hippocampus, amygdala, and hypothalamus), the prefrontal cortex (PFC), and the thalamus (Thal). Chronic exposure to stress is associated with an increased risk of anxiety disorders, depression, and dysregulation of sleep and feeding. (**B**). The endocannabinoid system plays a crucial neuroprotective role by mitigating oxidative stress and inhibiting excessive glutamatergic activity. Furthermore, endocannabinoids modulate synaptic plasticity, thereby facilitating the brain’s adaptation to changing environmental conditions.

**Figure 3 ijms-26-05929-f003:**
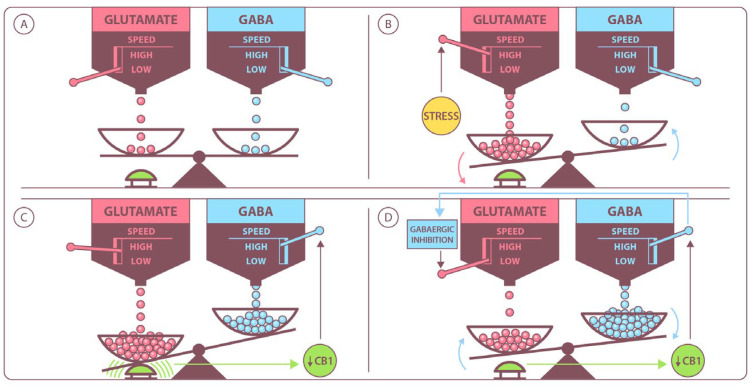
Model depicting the role of glutamine and gamma-aminobutyric acid (GABA) in stress adaptation mediated by cannabinoid type 1 (CB1) receptor activation. Among the various neurotransmitters involved in emotional processing, the glutamatergic (**left**) and GABAergic (**right**) systems play a key role in regulating anxiety. (**A**) Under normal conditions, the balance between excitatory and inhibitory transmission ensures appropriate emotional responses. (**B**) Stress disrupts the equilibrium between glutamatergic and GABAergic (excitatory–inhibitory) signaling. (**C**) CB1 receptors located on GABAergic terminals exhibit variable expression depending on the type of stimulus. Stress leads to the downregulation of CB1 receptors, altering the balance of CB1 receptor activation by endocannabinoids in the glutamatergic and GABAergic systems. (**D**) This prolonged downregulation of CB1 receptors on GABAergic terminals results in sustained GABAergic inhibition of glutamatergic transmission.

## Data Availability

All data and analysis are available within the manuscript or upon request to the corresponding author.

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
