# Peer review of "The Involvement of the Endocannabinoid, Glutamatergic, and GABAergic Systems in PTSD"

_ijms, 2025, doi:10.3390/ijms26135929_

Round 1
Reviewer 1 Report
Comments and Suggestions for Authors
The review of recent articles on the role of endocannabinoid, glutamatergic and GABAergic systems in post-traumatic stress disorders (PTSD) is of great interest in view of the increasing prevalence of PTSD and the search for valuable treatments. A better knowledge of the neurobiological systems involved that can be targeted would help in therapeutic strategies.
The 91 published articles cited are recent, with almost all of them published since 2020, and fit the purpose of the review, but the description of these articles is not adequate for this purpose. For instance, the author describes the glutamatergic and GABAergic systems in detail, while precisions on the relationship between these systems and the endocannabinoid system and PTSD is almost missing. Also missing is the report of top-down control of the memory engram in PTSD, which the author claims was a goal of the grant that supported this work. The author should better check the cited references as some sentences do not match, e.g. reference 42 describes corticosterone injection and GABA receptor modulators during fear extinction and not as written "CB1R activation in GABAergic neurons has been shown to reduce active coping strategies during fear conditioning in response to stress stimuli".
In addition, the author should describe the figure in detail in the figure legends (including the abbreviations used), but also in the text. For example, the role of cortisol in controlling inflammation and the relationship between hypocortisolism and neuroinflammation mentioned in Figure 1 was not described in the text. It needs to be explained how neuroinflammation is a key feature involved in PTSD. Again, the role of the hippocampus in PTSD is missing from the text following Figure 1.
The title of Figure 2 does not correspond to the Figure 2. Again, the effects under the umbrella of the endocannabinoid system mentioned in Figure 2 are not detailed afterwards. There is an error in Figure 2: the olfactory bulb is not part of the limbic system.
In summary, many parts of the review are off-topic. The manuscript needs to focus on the relationships between the four players of the review, i.e. the endocannabinoid system, PTSD, glutamatergic and GABAergic systems.
Author Response
Dear Editors,
In response to Reviewer 1’s comments, the following revisions have been made to the manuscript:
- Page 6 – Additional information has been included regarding the interaction between the endocannabinoid system and PTSD.
- Page 6 – New content has been added on top-down control of memory engrams in individuals with post-traumatic stress disorder (PTSD). The author's original research on this subject will be presented in a separate article.
- The reference in position 42 has been corrected.
- The legend of Figure 1 and the corresponding description in the text have been revised.
- The legend of Figure 2 and the corresponding description in the text have been revised.
- The relationships between the endocannabinoid system, PTSD, the glutamatergic system, and the GABAergic system have been described.
Sincerely,
The Author
Reviewer 2 Report
Comments and Suggestions for Authors
This review paper brings together evidence analyzing the roles of the endocannabinoid, glutamatergic, and GABAergic systems in the trauma process, with a focus on post-traumatic stress disorders (PTSD). The study utilized the PubMed and Web of Science databases, including articles published up to March 26, 2025. This work could be interesting for publication. However, some suggestions and issues could be improved. See below.
- Page 3 of 17 (lines 5-7; 10-15; 23-25). Please, provide references.
- Page 4 of 17. Line 1: references 29-31 refer to reviews. Can you provide and discuss single research studies? Lines 3-6: The author could provide additional references to support the sentence. Lines 10-11: The sentence is not consistent with the title of the review and the following analysis.
- Page 4 of 17. Can you discriminate between research articles and reviews used in this study?
- Page 5 of 17. Line 38-40, sentence on biomarkers. The author could clarify whether these results refer to animal or human studies.
- The chapter about the role of the glutamatergic system is exciting and thoroughly discussed. However, the role of this pathway in PTSD is only addressed in the last part of the chapter, leaving room instead for the detailed description of the glutamatergic system. I suggest developing the part about the interplay between the glutamate system and PTSD.
- About the role of the GABAergic system in PTSD, I have the same suggestion given above.
- In the abstract, the author states that there will be a focus on new therapeutic targets for trauma-related disorders. However, this aspect is not present in the text. Could you discuss it, or rephrase the aims of the study in the abstract?
- Check typos, please.
Author Response
Dear Editors,
In response to Reviewer 2’s comments, the following revisions have been made to the manuscript:
- Page 3 of 17 (lines 5–7; 10–15; 23–25): References have been added.
- Page 4 of 17:
- Line 1: References 29–31 – The cited studies have been discussed, and the references corrected.
- Lines 3–6: Additional references have been included.
- Lines 10–11: The sentence has been revised.
- Page 4 of 17: Relevant research articles have been cited.
- Page 5 of 17 (lines 38–40): It has been clarified that the discussed studies refer to human subjects.
- The section on the role of the glutamatergic system has been supplemented with information on its interaction with PTSD.
- The section on the role of the GABAergic system has been expanded to include its relationship with PTSD.
- The sentence regarding the aim of the study in the abstract has been revised.
- Typographical errors have been corrected.
Sincerely,
The Author
Round 2
Reviewer 1 Report
Comments and Suggestions for Authors
The authors revised the manuscript according to some comments from the reviewer. However, the sections on the glutamatergic and GABAergic systems still need to be rewritten, as some parts are off-topic. Additionally, parts of the section on the glutamatergic system were plagiarized from two published papers (Front Endocrinol. 2013 Oct 15;4:149. doi: 10.3389/fendo.2013.00149; Mol Brain. 2023 May 20;16:43. doi: 10.1186/s13041-023-01035-9). Finally, although some sentences have been added, the relationships between the endocannabinoid system, PTSD, and the glutamatergic and GABAergic systems require further development.
Reference 42 has not been corrected: “Specifically, CB1R activation in GABAergic neurons has been shown to reduce active coping strategies during fear conditioning in response to stress stimuli [42]”.
Author Response
Dear Editor,
Thank you very much for forwarding the comments from Reviewer 1 and for the opportunity to respond.
In response to the suggestions raised, the following changes have been made to the manuscript:
- In line with the Reviewer’s suggestion, the discussion regarding the relationship between the endocannabinoid system, PTSD, and the glutamatergic and GABAergic systems will be presented in a separate article, which is currently in preparation.
- The reference in position 42 has been corrected.
- After implementing the above-mentioned revisions, the author kindly requests that the remaining content of the manuscript be accepted without further modifications.
If the Reviewer believes that any part of the manuscript constitutes plagiarism, the author respectfully requests precise identification of the relevant sentences.
Sincerely,
The Author